# Template based Graph Neural Network with Optimal Transport Distances

**Cédric Vincent-Cuaz**
Univ. Côte d'Azur, INRIA, CNRS, LJAD
F-06100 Nice
`cedric.vincent-cuaz@inria.fr`

**Rémi Flamary**
IP Paris, CMAP, UMR 7641
F-91120 Palaiseau
`remi.flamary@polytechnique.edu`

**Marco Corneli**
Univ. Côte d'Azur, INRIA, CNRS, LJAD
F-06100 Nice
`marco.corneli@inria.fr`

**Titouan Vayer**
Univ. Lyon, INRIA, CNRS, ENS de Lyon
LIP UMR 5668, F-69342 Lyon
`titouan.vayer@inria.fr`

**Nicolas Courty**
Univ. Bretagne-Sud, CNRS, IRISA
F-56000 Vannes
`nicolas.courty@irisa.fr`

## Abstract

Current Graph Neural Networks (GNN) architectures generally rely on two important components: node features embedding through message passing, and aggregation with a specialized form of pooling. The structural (or topological) information is implicitly taken into account in these two steps. We propose in this work a novel point of view, which places distances to some learnable graph templates at the core of the graph representation. This distance embedding is constructed thanks to an optimal transport distance: the Fused Gromov-Wasserstein (FGW) distance, which encodes simultaneously feature and structure dissimilarities by solving a soft graph-matching problem. We postulate that the vector of FGW distances to a set of template graphs has a strong discriminative power, which is then fed to a non-linear classifier for final predictions. Distance embedding can be seen as a new layer, and can leverage on existing message passing techniques to promote sensible feature representations. Interestingly enough, in our work the optimal set of template graphs is also learnt in an end-to-end fashion by differentiating through this layer. After describing the corresponding learning procedure, we empirically validate our claim on several synthetic and real life graph classification datasets, where our method is competitive or surpasses kernel and GNN state-of-the-art approaches. We complete our experiments by an ablation study and a sensitivity analysis to parameters.

## 1 Introduction

Attributed graphs are characterized by *i)* the relationships between the nodes of the graph (structural or topological information) and *ii)* some specific features or attributes endowing the nodes themselves. Learning from those data is ubiquitous in many research areas [3], *e.g.* image analysis [21, 8], brain connectivity [30], biological compounds [23] or social networks [69], to name a few. Various methodologies approach the inherent complexity of those data, such as signal processing [54],

36th Conference on Neural Information Processing Systems (NeurIPS 2022).

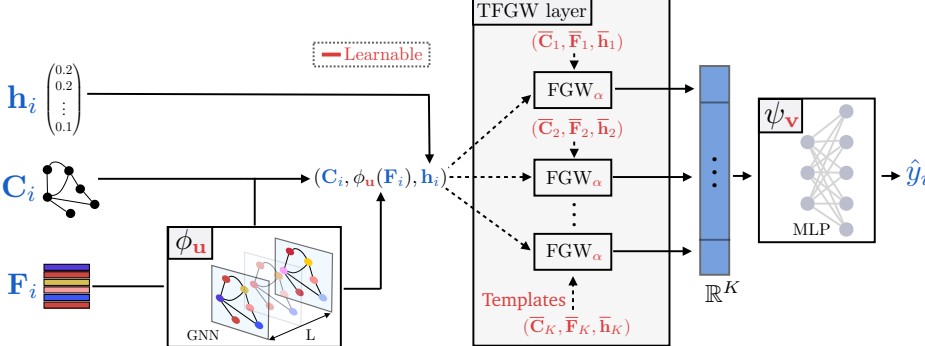

Figure 1: Illustration of the proposed model. **(left)** The input graph is represented as a triplet $(C_i, F_i, h_i)$ where the matrix $C_i$ encodes the structure, $F_i$ the features, $h_i$ the nodes' weights. A GNN $\phi_u$ is applied to the raw features in order to extract a meaningful node representations. **(center)** The TFGW layer is applied to the filtered graph and provides a vector representation as FGW distances to templates. **(right)** a final MLP $\psi_v$ is applied to this vector in order to predict the final output of the model. All objects in red are parameters that are learned from the data.

Bayesian and kernel methods on graphs [41, 29] or more recently Graph Neural Networks (GNN) [64] in the framework of the geometric deep learning [8, 7].

We are interested in this work in the classification of attributed graphs **at the instance level**. One existing approach consists in designing kernels that leverage topological properties of the observed graphs [5, 17, 19, 53]. For instance, the popular Weisfeiler-Lehman (WL) kernel [52] iteratively aggregates for each node the features of its $k$-hop neighborhood. Alternative approaches aim at learning vectorial representations of the graphs that can encode the graph structure (*i.e. graph representation learning* [10]). In this domain, GNN lead to state-of-the-art performances with end-to-end learnable embeddings [64]. At a given layer, these architectures typically learn the node embeddings via local permutation-invariant transformations aggregating its neighbour features [37, 25, 20, 67]. In order to obtain a representation of the whole graph suitable for classification, GNNs finally operate a pooling [26, 40] of the node embeddings, either global (*e.g* summation over nodes [67]), or hierarchical (*e.g* by iteratively clustering nodes [71, 70, 32]).

Another line of works targets the construction of meaningful distances that integrate simultaneously the structural and feature information, and that are based on optimal transport (OT) [61, 46]. Originally designed to compare probability distributions based on a geometric notion of optimality, it allows defining very general loss functions between various objects, modeled as probability distributions. In a nutshell, it proceeds by constructing a *coupling* between the distributions that minimizes a specific *cost*. Some approaches dealing with graphs rely on non-parametric models that first embed the graphs into a vectorial space and then match them via OT [43, 56, 27, 35]. Recently [11] proposed the OT-GNN model, that embeds a graph as a vector of the Wasserstein distances between the nodes' embeddings (after GNN pre-processing) and learnt point clouds, acting as templates.

Building further from OT variants, the Gromov-Wasserstein (GW) distance [39] directly handles graphs through the symmetric matrix $C$ that encodes the distance/similarity between every pairs of nodes (*e.g.* adjacency, shortest path) and a *weight vector* $h$ on the nodes encoding the nodes' relative importance. GW has proven to be useful for tasks such as graph matching and partitioning [66, 15] or unsupervised graph dictionary learning [65, 63, 62]. GW has been also extended to directed graphs [14] and to attributed graphs via the Fused Gromov-Wasserstein (FGW) distance [59, 58], that realizes a trade-off between an OT distance with a cost on node features and the GW distance between the similarity matrices. Despite its recent successes on complex unsupervised tasks such as graph clustering [65, 63, 62], FGW has never been explored as part of an end-to-end model for graph classification. In this work, we fill this gap by introducing a novel "layer" that embeds an attributed graph into a vector, whose coordinates are FGW distances to few (learned) graph templates. While FGW can be performed directly on raw data (*i.e.* the input structured graph without any pre-processing), we also consider the case where features representations are learnt from a GNN, similarly to OT-GNN [11], and thus also realizing a particular type of aggregation.

**Contributions.** We introduce a new GNN layer, named **TFGW** for **Template-based FGW** and illustrated in the center of Figure 1. From an input graph, it computes a vector of FGW distances to learnable graph templates. This layer that can be seen as an alternative to global pooling layers and can be integrated into any neural network architecture. We discuss its properties and the associated invariances. We detail the optimization strategy that enables learning simultaneously GNN pre-processing layers and graph templates relevant for a downstream task in an end-to-end fashion. We empirically demonstrate the relevance of our model in terms of performances compared to several state-of-the-art architectures. Remarkably, we show that a simple GNN model leveraging on our new layer can surpass state-of-the-art performances by a relatively large margin. Finally, we also provide some illustrative interpretations of our method and a sensitivity analysis of our model parameters.

## 2 Fused Gromov-Wasserstein template based layer

In order to describe our novel template-based GNN layer we first introduce more formally the FGW distance and its properties. In the following we denote by $\Sigma_n := \{\boldsymbol{h} \in \mathbb{R}_+^n | \sum_i h_i = 1\}$ the probability simplex with $n$-bins, and by $\mathbb{S}_n(\mathbb{A})$ the set of symmetric matrices of size $n$ taking values in $\mathbb{A} \subset \mathbb{R}$.

### 2.1 Fused Gromov-Wasserstein distance

An undirected attributed graph $\mathcal{G}$ with $n$ nodes can be modeled in the OT context as a tuple $(\boldsymbol{C}, \boldsymbol{F}, \boldsymbol{h})$, where $\boldsymbol{C} \in \mathbb{S}_n(\mathbb{R})$ is a matrix encoding relationships between nodes, $\boldsymbol{F} = (\boldsymbol{f}_1, ..., \boldsymbol{f}_n)^\top \in \mathbb{R}^{n \times d}$ is a node feature matrix and $\boldsymbol{h} \in \Sigma_n$ is a vector of weights modeling the relative importance of the nodes within the graph (Figure 1, left). Without any prior knowledge, uniform weights can be chosen ($\boldsymbol{h} = \boldsymbol{1}_n/n$). The matrix $\boldsymbol{C}$ can be the graph adjacency matrix, the shortest-path matrix or any other description of the node relationships (i.e. the topology) of the graph [47, 58, 15]. Let us now consider two such graphs $(\boldsymbol{C}, \boldsymbol{F}, \boldsymbol{h})$ and $(\overline{\boldsymbol{C}}, \overline{\boldsymbol{F}}, \overline{\boldsymbol{h}})$, of respective sizes $n$ and $\overline{n}$ (with possibly $n \neq \overline{n}$). The Fused Gromov-Wasserstein (FGW) distance is defined for $\alpha \in [0, 1]$ as [58, 59]:

$$\text{FGW}_\alpha(\boldsymbol{C}, \boldsymbol{F}, \boldsymbol{h}, \overline{\boldsymbol{C}}, \overline{\boldsymbol{F}}, \overline{\boldsymbol{h}}) = \min_{\boldsymbol{T} \in \mathcal{U}(\boldsymbol{h}, \overline{\boldsymbol{h}})} \sum_{ijkl} \left( \alpha(C_{ij} - \overline{C}_{kl})^2 + (1-\alpha)\|\boldsymbol{f}_i - \overline{\boldsymbol{f}}_k\|_2^2 \right) T_{ik}T_{jl} \quad (1)$$

where $\mathcal{U}(\boldsymbol{h}, \overline{\boldsymbol{h}}) := \{\boldsymbol{T} \in \mathbb{R}_+^{n \times \overline{n}} | \boldsymbol{T}\boldsymbol{1}_{\overline{n}} = \boldsymbol{h}, \boldsymbol{T}^\top \boldsymbol{1}_n = \overline{\boldsymbol{h}}\}$ is the set of admissible coupling between $\boldsymbol{h}$ and $\overline{\boldsymbol{h}}$. FGW aims at finding an optimal coupling $\boldsymbol{T}^\star$ by minimizing a trade-off cost, via $\alpha$, between a Wasserstein (W) cost on the features and a Gromov-Wasserstein (GW) cost on the similarity matrices, both sharing the same coupling. The optimal coupling $\boldsymbol{T}^\star$ acts as a soft matching of the nodes, which tends to associate pairs of nodes that have similar pairwise relations in $\boldsymbol{C}$ and $\overline{\boldsymbol{C}}$ (GW cost), and similar features in $\boldsymbol{F}$ and $\overline{\boldsymbol{F}}$ (W cost).

Interestingly, FGW defines a metric on the space of attributed graphs. In particular, if $\boldsymbol{C}$ and $\overline{\boldsymbol{C}}$ are shortest-path matrices, the FGW distance vanishes if and only if the two attributed graphs are the same up to a permutation [59, Theorem 3.2]. Such invariance involves that two graphs *strongly* isomorphic according to Weisfeiler-Lehman base tests [33, 56] will have a zero FGW distance for any $\alpha$ and, more importantly, $\text{FGW}_\alpha = 0$ implies that the graphs are strongly isomorphic[1]. When $\boldsymbol{C}, \overline{\boldsymbol{C}}$ are any symmetric matrices, we can mention that GW ($\alpha = 1$) also defines a pseudo-distance [55, Theorem 5.8] with respect to the notion of *weak* isomorphism [55, 15].

**Solving for FGW.** The optimization problem 1 is a non-convex quadratic program [59, equation 6], whose non-convexity comes from the GW cost. A possible optimization procedure to solve this problem is a Conditional Gradient (CG) algorithm, which is known to converge to a local optimum [31]. The computational complexity of each iteration is $O(n^2\overline{n} + \overline{n}^2n)$ [47]. Thus, if two graphs of considerably different sizes are considered, the complexity is quadratic with respect to the largest size. Existing attempts to reduce this computational cost either exploit entropic regularization of OT [47, 50] or graph partitioning [66, 13].

---

[1]Two graphs $(\boldsymbol{C}, \boldsymbol{F}, \boldsymbol{h})$ and $(\overline{\boldsymbol{C}}, \overline{\boldsymbol{F}}, \overline{\boldsymbol{h}})$ are strongly isomorphic if $n = \overline{n}$ and there exists a permutation matrix $\mathbf{P} \in \{0,1\}^{n \times n}$ such that $\overline{\boldsymbol{C}} = \mathbf{P}\boldsymbol{C}\mathbf{P}^\top, \overline{\boldsymbol{F}} = \mathbf{P}\boldsymbol{F}$ and $\overline{\boldsymbol{h}} = \mathbf{P}\boldsymbol{h}$

## 2.2 Template-based (T)FGW Graph Neural Networks

Building upon the FGW distance and its properties, we propose a simple layer for a GNN that takes a graph $(\boldsymbol{C}, \boldsymbol{F}, \boldsymbol{h})$ as input and computes its FGW distances to a list of $K$ *template graphs* $\overline{\mathcal{G}} := \{(\overline{\boldsymbol{C}}_k, \overline{\boldsymbol{F}}_k, \overline{\boldsymbol{h}}_k)\}_{k \in [\![K]\!]}$ as follows :

$$\text{TFGW}_{\overline{\mathcal{G}}, \alpha}(\boldsymbol{C}, \boldsymbol{F}, \boldsymbol{h}) := \left[\text{FGW}_\alpha(\boldsymbol{C}, \boldsymbol{F}, \boldsymbol{h}, \overline{\boldsymbol{C}}_k, \overline{\boldsymbol{F}}_k, \overline{\boldsymbol{h}}_k)\right]_{k=1}^K \tag{2}$$

We postulate that this graph representation can be discriminant between the observed graphs due to FGW. This claim relies on the theory of [2] allowing one to learn provably strongly discriminant classifiers based on the distances from the observed graphs and templates that are sampled from the dataset (see e.g. [48] adopting the Wasserstein distance). However such an approach often requires a large amount of templates which might be prohibitive if the distance is costly to compute. Instead, we propose to **learn** the graph templates $\overline{\mathcal{G}}$ in a supervised manner. In the same way, we also learn the trade-off parameter $\alpha$ on the data. As such, the TFGW layer can automatically adapt to the data whose discriminating information can be discovered either in the features or in the structure of the graphs, or in a combination of the two. Moreover, the template structures can leverage on any type of input representation $\boldsymbol{C}_i$ since they are learnt directly from the data. Indeed, in the numerical experiments we implemented the model using either adjacency matrices (ADJ) that provide more interpretable templates (component $C_{i,j} \in [0, 1]$ can be seen as a probability of link between nodes) or shortest path matrices (SP) that are more complex to interpret but encode global relations between the nodes.

The TFGW layer can be used directly as a first layer to build a graph representation feeding a fully connected network (MLP) for *e.g.* graphs classication. In order to enhance the discriminating power of the model, we propose to put a GNN (denoted by $\phi_{\boldsymbol{u}}$ and parametrized by $\boldsymbol{u}$) on top of the TFGW layer. We assume in the remainder that this GNN model $\phi_{\boldsymbol{u}}$ is injective in order to preserve isomorphism relations between graphs (see [67] for more details). With a slight abuse of notation, we write $\phi_{\boldsymbol{u}}(\mathbf{F})$ to denote the feature matrix of an observed graph after being processed by the GNN.

**Learning with TFGW-GNN.** We focus on a classification task where we observe a dataset $\mathcal{D}$ of $I$ graphs $\{\mathcal{G}_i = (\boldsymbol{C}_i, \boldsymbol{F}_i, \boldsymbol{h}_i)\}_{i \in [\![I]\!]}$ with variable number of nodes $\{n_i\}_{i \in [\![I]\!]}$ and where each graph is assigned to a label $y_i \in \mathcal{Y}$, with $\mathcal{Y}$ a finite set. The full model is illustrated in Figure 1. We first process the features of the nodes of the input graphs via the GNN $\phi_{\boldsymbol{u}}$, then use the TFGW layer to represent the graphs as vectors in $\mathbb{R}^K$. Finally we use the final MLP model $\psi_{\boldsymbol{v}} : \mathbb{R}^K \to \mathcal{Y}$ parameterized by $\boldsymbol{v}$, to predict the label for any input graph. The whole model is learned in a end-to-end fashion by minimizing the cross-entropy loss on the whole dataset leading to the following optimization problem :

$$\min_{\boldsymbol{u}, \boldsymbol{v}, \{(\overline{\boldsymbol{C}}_k, \overline{\boldsymbol{F}}_k, \overline{\boldsymbol{h}}_k)\}, \alpha} \quad \frac{1}{I} \sum_{i=1}^I \mathcal{L}\left(y_i, \psi_{\boldsymbol{v}}\left(\text{TFGW}_{\overline{\mathcal{G}}, \alpha}(\boldsymbol{C}_i, \phi_{\boldsymbol{u}}(\boldsymbol{F}_i), \boldsymbol{h}_i)\right)\right). \tag{3}$$

Notable parameters of (3) are the template graphs in the embeddings $\{(\overline{\boldsymbol{C}}_k, \overline{\boldsymbol{F}}_k, \overline{\boldsymbol{h}}_k)\}$ and more precisely their pairwise node relationship $\overline{\boldsymbol{C}}_k$, node features $\overline{\boldsymbol{F}}_k$ and the distribution on the nodes $\overline{\boldsymbol{h}}_k$ on the simplex. The last parameter reweighs individual nodes in each template and performs nodes selection when some weights are exactly 0 [63, 62]. Finally, the global parameter $\alpha$ is also learnt from the whole dataset. Although it is possible to learn a different $\alpha$ per template, we observed that this extra level of flexibility is prone to overfitting, and we will not consider it in the experimental section.

**Optimization and differentiation of TFGW.** We propose to solve the optimization problem in (3) using stochastic gradient descent. The FGW distances are computed by adapting the conditional gradient solver implemented in the POT toolbox [18]. The solver was designed to allow backward propagation of the gradients *w.r.t.* all the parameters of the distance and was adapted to also compute the gradient *w.r.t.* the parameter $\alpha$. The gradients are obtained using the Envelop Theorem [1] allowing to keep $\boldsymbol{T}^\star$ constant. We used Pytorch [45] to implement the model. The template structure $\overline{\boldsymbol{C}}_k$, node weights $\overline{\boldsymbol{h}}_k$ and $\alpha$ are updated with a projected gradient respectively on the set of symmetric matrices $\mathbb{S}_{\overline{n}_k}(\mathbb{R}_+)$ ($\mathbb{S}_{\overline{n}_k}([0, 1])$ when $\boldsymbol{C}_i$ are adjacency matrices), the simplex $\Sigma_{\overline{n}_k}$ and $[0, 1]$. The projection onto the probability simplex of the node weights leads to sparse solutions [16], therefore

the size of each $(\overline{C}_k, \overline{F}_k, \overline{h}_k)$ can decrease along iterations hence reducing the *effective* number of their parameters to optimize. This way the numerical solver can leverage on the fact that many computations are unnecessary as soon as the weights are set to zero. Note that the FGW solver from POT uses an OT solver implemented in C++ on CPU which means that it comes with some overhead (memory transfer between GPU and CPU) when training the model on GPU. Still the multiple FGW distances computation has been implemented in parallel on CPU with a computational time that remains reasonable in practice (see experimental section 3.3). While a GPU solver can be found when using entropy regularized FGW, it introduces a new parameter related to the regularization strength which is more cumbersome to set, and that we did not consider it in the experiments.

**Properties of the TFGW layer.** We now discuss a property of the proposed layer resulting from the properties of FGW (see Section 2.1). We have the following result:

**Lemma 1** *The* TFGW *embeddings are invariant to strong isomorphism.*

This lemma directly stems from the fact that FGW is invariant to strong isomorphism of one of its inputs. This proposition implies that two graphs with any aforementioned representation which only differ by a permutation of the nodes will share the same TFGW embedding. Moreover such a property holds for any mapping $\phi_u$ which is injective, such as a Multi-Layer Perceptron (MLP) [22] or any GNN with a sum aggregation scheme as described in [67].

Moreover, the optimal coupling $T^\star$ resulting from TFGW between $(C_i, \phi_u(F_i), h_i)$ and the template $(\overline{C}_k, \overline{F}_k, \overline{h}_k)$, will encode correspondances between the nodes of the graph and the nodes of the template that will be propagated during the backward operation. The size of the inputs, the size of the templates and their respective weight $\overline{h}_k$ will play a crucial role regarding this operation. Also note that, since the templates are estimated here to optimize a supervised task, they will promote discriminant distance embedding instead of graph reconstruction quality as proposed in other FGW unsupervised learning methods [63, 62].

## 3 Numerical experiments

This section aims at illustrating the performances of our approach for graph classification in synthetic and real-world datasets. First, we showcase the relevance of our TFGW layer on existing synthetic datasets known to require expressiveness beyond the WL-test (Section 3.1). Then we benchmark our model with state-of-the-art approaches on well-known real-world datasets (Section 3.2). We finally discuss our results through a sensitivity analysis of our models (Section 3.3).[2]

### 3.1 Synthetic datasets beyond WL test

Identification of graphs beyond the WL test is one important challenge faced by the GNN community. In order to test the ability of TFGW to handle such fundamentally difficult problems we consider two synthetic datasets: 4-CYCLES [34, 44] contains graphs with (possibly) disconnected cycles where the label $y_i$ is the presence of a cycle of length 4; SKIP-CIRCLES [12] contains cir-

Table 1: Average accuracy on synthetic datasets (10 simulations).

| model | 4-CYCLES | SKIP-CIRCLES |
|---|---|---|
| TFGW | **0.99(0.03)** | **1.00(0.00)** |
| TFGW-fix | 0.63(0.11) | **1.00(0.00)** |
| GIN | 0.50(0.00) | 0.10(0.00) |
| DropGIN | **1.00(0.01)** | 0.82(0.28) |

cular graphs with skip links and the labels (10 classes) are the lengths of the skip links among $\{2, 3, 4, 5, 6, 9, 11, 12, 13, 16\}$.

We compare the performances of the TFGW layer for embedding such graphs with GIN [67] designed to be at least as expressive as the WL test, and DropGIN [44] which proposed a successful dropout technique to overcome some drawbacks of GIN. We replicate the benchmark of [44] by considering for both GIN and DropGIN, 4 GIN layers for 4-CYCLES, and 9 GIN layers for SKIP-CIRCLES as the skip links can form cycles of up to 17 hops. Since the graphs do not have features we use directly the TFGW on the raw graph representation with $\alpha = 1$ hence computing only the GW distance. The GNN methods above artificially adds a feature equal to 1 on all nodes as they have the same degree. For these experiments we use adjacency matrices for $C_i$ and we

---

[2]Code available at `https://github.com/cedricvincentcuaz/TFGW`.

Table 2: Test set classification accuracies from 10-fold CV. The first (resp. second) best performing method is highlighted in bold (resp. underlined).

| category | model | MUTAG | PTC | ENZYMES | PROTEIN | NCI1 | IMDB-B | IMDB-M | COLLAB |
|---|---|---|---|---|---|---|---|---|---|
| Ours | TFGW ADJ (L=2) | **96.4(3.3)** | **72.4(5.7)** | 73.8(4.6) | **82.9(2.7)** | **88.1(2.5)** | **78.3(3.7)** | **56.8(3.1)** | **84.3(2.6)** |
| | TFGW SP (L=2) | 94.8(3.5) | 70.8(6.3) | **75.1(5.0)** | 82.0(3.0) | 86.1(2.7) | 74.1(5.4) | 54.9(3.9) | 80.9(3.1) |
| OT emb. | OT-GNN (L=2) | 91.6(4.6) | 68.0(7.5) | 66.9(3.8) | 76.6(4.0) | 82.9(2.1) | 67.5(3.5) | 52.1(3.0) | 80.7(2.9) |
| | OT-GNN (L=4) | 92.1(3.7) | 65.4(9.6) | 67.3(4.3) | 78.0(5.1) | 83.6(2.5) | 69.1(4.4) | 51.9(2.8) | 81.1(2.5) |
| | WEGL | 91.0(3.4) | 66.0(2.4) | 60.0(2.8) | 73.7(1.9) | 75.5(1.4) | 66.4(2.1) | 50.3(1.0) | 79.6(0.5) |
| GNN | PATCHYSAN | 91.6(4.6) | 58.9(3.7) | 55.9(4.5) | 75.1(3.3) | 76.9(2.3) | 62.9(3.9) | 45.9(2.5) | 73.1(2.7) |
| | GIN | 90.1(4.4) | 63.1(3.9) | 62.2(3.6) | 76.2(2.8) | 82.2(0.8) | 64.3(3.1) | 50.9(1.7) | 79.3(1.7) |
| | DropGIN | 89.8(6.2) | 62.3(6.8) | 65.8(2.7) | 76.9(4.3) | 81.9(2.5) | 66.3(4.5) | 51.6(3.2) | 80.1(2.8) |
| | PPGN | 90.4(5.6) | 65.6(6.0) | 66.9(4.3) | 77.1(4.0) | 82.7(1.8) | 67.2(4.1) | 51.3(2.8) | 81.0(2.1) |
| | DIFFPOOL | 86.1(2.0) | 45.0(5.2) | 61.0(3.1) | 71.7(1.4) | 80.9(0.7) | 61.1(2.0) | 45.8(1.4) | 80.8(1.6) |
| Kernels | FGW - ADJ | 82.6(7.2) | 55.3(8.0) | 72.2(4.0) | 72.4(4.7) | 74.4(2.1) | 70.8(3.6) | 48.9(3.9) | 80.6(1.5) |
| | FGW - SP | 84.4(7.3) | 55.5(7.0) | 70.5(6.2) | 74.3(3.3) | 72.8(1.5) | 65.0(4.7) | 47.8(3.8) | 77.8(2.4) |
| | WL | 87.4(5.4) | 56.0(3.9) | 69.5(3.2) | 74.4(2.6) | 85.6(1.2) | 67.5(4.0) | 48.5(4.2) | 78.5(1.7) |
| | WWL | 86.3(7.9) | 52.6(6.8) | 71.4(5.1) | 73.1(1.4) | 85.7(0.8) | 71.6(3.8) | 52.6(3.0) | 81.4(2.1) |
| | Gain with TFGW | **4.3** | **4.4** | **2.9** | **4.9** | **2.4** | **5.3** | **4.2** | **2.9** |

investigate two flavours of TFGW: 1) in TFGW-fix we fix the templates by sampling *one template per class* from the training dataset (this can be seen as a simpler FGW feature extraction); 2) for TFGW we learn the templates from the training data (as many as the number of classes) as proposed in the previous sections. Results are averaged over 10 runs and reported in Table 1. TFGW based methods perform very well on both datasets with impressive results on SKIP-CIRCLE when GNN have limited performances. This is due to the fact that different samples from one class of SKIP-CIRCLE are generated by permuting nodes of the same graph and FGW distances are invariant to these permutations. 4-CYCLES has a more complex structure with intra-class heterogeneity and requires more than two templates to perform as good as DropGIN. To illustrate this we have computed the accuracy on this dataset as a function of the number of templates $K$ in Figure 2. We can see that a perfect classification is reached up to $K = 4$ for TFGW, while TFGW-fix still struggles to generalize at $K = 20$. This illustrates that learning the templates is essential to keep $K$ (and numerical complexity) small while ensuring good performances.

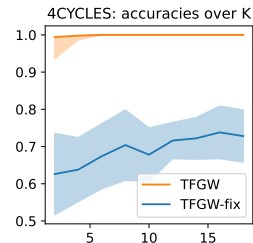

Figure 2: Test accuracy distributions by number of templates either fixed or learned.

## 3.2 Graph classification benchmmark

We now evaluate and compare the performances of our TFGW GNN with a number of state-of-the-art graph classifiers, from kernel methods to GNN. The numerical experiments are conducted on real life graph datasets to provide a fair benchmark of all methods on several heterogeneous graph structures.

**Datasets.** We use 8 well-known graph classification datasets [24]: 5 bioinformatics datasets among which 3 have discrete node features (MUTAG, PTC, NCI1 [28, 52]) and 2 have continuous node features (ENZYMES, PROTEINS[6]) and 3 social network datasets (COLLAB, IMDB-B, IDBM-M [69]). In order to analyse them with all methods, we augment unattributed graphs from social networks with node degree features. Detailed description and statistics on these datasets are reported in the supplementary material.

**Baselines.** We benchmark our approaches to the following state-of-the-art baselines for graphs classification, split into 3 categories: i) *kernel based approaches*, including FGW [59] operating on adjacency and shortest-path matrices, the WL subtree kernel [52, WL] and the Wasserstein WL kernel [56, WWL]. For these methods that do not require a stopping criterion dependent on a validation set, we report results using for parameter validation a 10-fold nested cross-validation [59, 29] repeated 10 times. ii) *OT based representation learning* models, including WEGL [27] and OT-GNN [11]. iii) *GNN models*, with global or more sophisticated pooling operations, including PATCHY-SAN [42], DIFFPOOL [70], PPGN [36], GIN [67] and its augmented version through structure perturbations DropGIN [44]. For all these methods, we adopt the hyper-parameters suggested in the respective papers, but with a slightly different model selection scheme, as detailed in the next paragraph.

**Benchmark settings.** Recent GNN literature [37, 67, 36, 44] successfully addressed many limitations in terms of model expressiveness compared to the WL tests. Within that scope, they suggested to benchmark their models using a 10-fold cross-validation (CV) where the best average accuracy on the validation folds was reported. We suggest here to quantify the generalization capacities of GNN based models by performing a 10-fold cross validation with a holdout test set never seen during training. For each split, we track the accuracy on the validation fold every 5 epochs, then the model whose parameters maximize that accuracy is retained. Finally, the model used to predict on the holdout test set is the one with maximal validation accuracy averaged across all folds. This setting is more realistic than a simple 10-fold CV and allows a better understanding of the generalization performances [4]. This point explains why some existing approaches have here different performances than those reported in their original paper.

For all the TFGW based approaches we empirically study the impact of the input structure representation by considering adjacency (ADJ) and shortest-path (SP) matrices $C_i$. For all template based models, we set the size of the templates to the median size of the observed graphs.

We validate the number of templates $K$ in $\{\beta|\mathcal{Y}|\}_\beta$, with $\beta \in \{2, 4, 6, 8\}$ and $|\mathcal{Y}|$ the number of classes. Only for ENZYMES with 6 classes of 100 graphs each, we validate $\beta \in \{1, 2, 3, 4\}$. All parameters of our TFGW layers highlighted in red in Figure 1 are learned while $\phi_{\boldsymbol{u}}$ is a GIN architecture [67] composed of $L = 2$ layers aggregated using the Jumping Knowledge scheme [68] known to prevent overfitting in global pooling frameworks. For OT-GNN we validate the number of GIN layers in $L \in \{2, 4\}$. Finally for fairness, we validate the number of hidden units within the GNN layers and the application of dropout on the final MLP for predictions, similarly to GIN and DropGIN.

**Results analysis.** The results of the comparisons in terms of accuracy are reported in Table 2. Our TFGW approach consistently *outperforms with significant margins* the state-of-the-art approaches from all categories. Even if most of the benchmarked models can perfectly fit the train sets by learning implicitly the graphs structure [67, 44, 36], enforcing such knowledge explicitly as our TFGW layer does (through FGW distances) leads to considerably stronger generalization performances. On 7 out of 8 datasets, TFGW leads to better performances while operating on adjacency matrices (TFGW ADJ) than on shortest-path ones (TFGW SP). Interestingly, this ranking with respect to those input representations does not necessarily match the one of the FGW kernel which extracts knowledge from the graph structures $C_i$ through FGW, as our TFGW layer. These different dependencies to the provided inputs may be due to the GNN pre-processing of node features which suggests the study of its ablation.

Finally to complete this analysis, we report in Table 3 the number of parameters of best selected models across various methods, for the dataset PTC. We refer the reader interested in an extended benchmark to the supplementary material. Our TFGW leads to better classification performances while having comparable number of parameters than these competitors. We also reported for these models, their averaged prediction time per graph. These measures were taken on CPUs (Intel Core i9-9900K CPU, 3.60 GHz) in order to fairly compare the numerical com-

Table 3: Number of parameters and averaged prediction time per graph.

| model | PTC | |
|---|---|---|
| | parameters | runtimes (ms) |
| (ours) TFGW | 25.1k | 12.1 |
| OT-GNN | 30.8k | 7.6 |
| GIN | 29.9k | 0.19 |
| DropGIN | 44.1k | 14.3 |

plexity of these methods, as OT solver used in TFGW and OT-GNN are currently limited to these devices (see the detailed discussion in Section 2.1). Although the theoretical complexity of our approach is *at most* cubic in the number of nodes, we still get in practice a fairly good speed for classifying graphs, in comparison to the other competitive methods.

### 3.3 Ablation study, sensitivity analysis and discussions

In this section we inspect the role of some of the model parameters ($\alpha$ in FGW, weights estimation in the templates, depth of the GNN $\phi_{\boldsymbol{u}}$) in terms of the classification performance. To this end, we first conduct on all datasets an ablation study on the graph template weights $\overline{\boldsymbol{h}}_k$ and the number of GIN layers in $\phi_{\boldsymbol{u}}$. Then, we take a closer look at the estimated trade-off parameters $\alpha$ and provide a sensitivity analysis *w.r.t.* the number of templates and the number of GIN layers.

**Ablation Study.** Following the same procedure as in Section 3.2, we benchmark the following settings for our TFGW models: for adjacency (ADJ) and shortest path (SP) representations $C_i$, we

Table 4: Classification results from 10-fold cross-validation of our TFGW models in various scenarios: for $L \in \{0, 1, 2\}$ GIN layers, we either fix templates weights $\overline{h}_k$ to uniform distributions or learn them. The first and second best performing method are respectively highlighted in bold and underlined.

| model | inputs | $\overline{h}_k$ | MUTAG | PTC | ENZYMES | PROTEIN | NCI1 | IMDB-B | IMDB-M | COLLAB |
|---|---|---|---|---|---|---|---|---|---|---|
| TFGW (L=0) | ADJ | uniform | 92.1(4.5) | 63.6(5.0) | 67.4(7.3) | 78.0(2.0) | 80.3(1.5) | 69.9(2.5) | 49.7(4.1) | 78.7(3.1) |
| | ADJ | learnt | 94.2(3.0) | 64.9(4.1) | 72.1(5.5) | 78.8(2.2) | 82.1(2.5) | 71.3(4.3) | 52.3(2.5) | 80.9(2.7) |
| | SP | uniform | 94.8(3.7) | 66.5(6.7) | 72.7(6.9) | 77.5(2.4) | 79.6(3.7) | 68.1(4.4) | 48.3(3.6) | 78.4(3.4) |
| | SP | learnt | 95.9(4.1) | 67.9(5.8) | 75.1(5.6) | 79.5(2.9) | 83.9(2.0) | 72.6(3.1) | 53.1(2.5) | 79.8(2.5) |
| TFGW(L=1) | ADJ | learnt | 94.8(3.1) | 68.7(5.8) | 72.7(5.1) | 81.5(2.8) | 85.4(2.8) | 76.3(4.3) | 55.9(2.4) | 82.6(1.8) |
| | SP | learnt | 95.4(3.5) | 70.9(5.5) | 74.9(4.8) | 82.1(3.4) | 85.7(3.1) | 73.8(4.8) | 54.2(3.3) | 81.1(2.5) |
| TFGW (L=2) | ADJ | learnt | 96.4(3.3) | 72.4(5.7) | 73.8(4.6) | 82.9(2.7) | 88.1(2.5) | 78.3(3.7) | 56.8(3.1) | 84.3(2.6) |
| | SP | learnt | 94.8(3.5) | 70.8(6.3) | 75.1(5.0) | 82.0(3.0) | 86.1(2.7) | 74.1(5.4) | 54.9(3.9) | 80.9(3.1) |

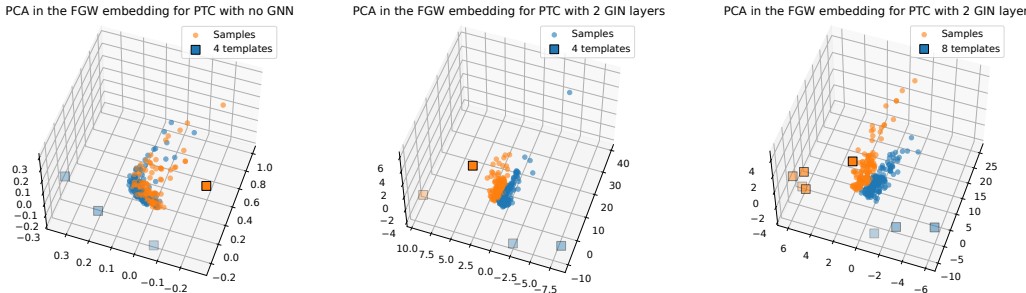

Figure 3: PCA projections of the template based embeddings for different models and number of templates.(need to debug colors)

learn the distance layers either directly, on the raw data (*i.e.* $L = 0$, $\phi_\mathbf{u} = \mathrm{id}$), or after embedding the data with ($L = 1$) or ($L = 2$) GIN layers. For $L = 0$ we either fix the graph template weights $\overline{h}_k$ uniformly or learn them. The results (test accuracy) are reported in Table 4. Learning the weights systematically improves the generalization capabilities of our models of at least $1\%$ or $2\%$ for both ADJ and SP graph representations. For a given number of graph templates, the weights learning allows to better fit the specificities of the classes (e.g. varying proportion of nodes in different parts of the graphs). Moreover, as weights can become sparse in the simplex during training they also allow the model to have templates whose number of nodes adapts to the classification objective, while bringing computational benefits as discussed in Section 2.2. Those observations explain why we learnt them by default in the benchmark of the previous subsection.

Next, we see in Table 4 that using GNN layers as a pre-processing for our TFGW layer enhances generalization powers of our models, whose best performances are obtained for $L = 2$. Interestingly, for $L = 0$, TFGW with SP matrices outperforms TFGW with ADJ matrices, meaning that the shortest path distance brings more discriminant information on raw data. But when $L \geq 1$ (*i.e.* when a GNN pre-processes the node features), TFGW with ADJ matrices improves the accuracy. An explanation could be that the GNN $\phi_\mathbf{u}$ can somehow replicate (and outperform) a SP metric between nodes. This emphasizes that the strength of our approach clearly exhibited in Table 2 lies in the inductive bias of our FGW distance embedding. This last point is further reinforced by additional experiments reported in the supplementary material, where GIN layers (used by default for our TFGW model) are replaced by GAT layers [60].

**Importance of the structure/feature aspect of FGW.** To the best of our knowledge we are the first to actually learn the trade-off parameter $\alpha$ of FGW in a supervised way. For this matter, we verify that our models did not converge to degenerated solutions where either the structure ($\alpha = 0$) or the features ($\alpha = 1$) are omitted. To this end we report in Figure 4 the distributions of the estimated $\alpha$ for some models learnt on datasets PTC and IMDB-B, where features are respectively existing in the dataset or created using

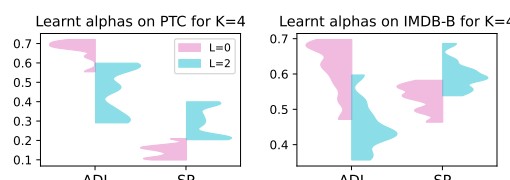

Figure 4: Distributions of estimated $\alpha$.

node degrees. We can see that for both kinds of input graph representations, $\alpha$ parameters are strictly between $0$ and $1$. One can notice the variances of those distributions illustrating the non-uniqueness of this trade-off parameter coming from the non-convexity of our optimization problem (a given value of $\alpha$ can potentially be compensated by the scaling of the GNN output). Unfortunately, the analysis of the real relative importance between structures and features can not be achieved only by looking at those values as the node embeddings and templates are different across models and data splits.

**Visualizing the TFGW embedding.** In order to interpret the TFGW embedding, we illustrate in Figure 3 the PCA projection of our distance embeddings learned on PTC with $L = 0$ and $L = 2$ and the number of templates $K$ varying in $\{4, 8\}$. For this experiment, we have chosen the PCA because it allows to have a more interpretable low dimensional projection that preserves the geometry compared to local neighbourhood based embeddings such as TSNE [57] or UMAP [38]. As depicted in the figure, the learned templates are extreme points in the embedding space of the PCA. This result is particularly interesting because existing unsupervised FGW representation learning methods tend toward estimating templates that belong to the data manifold, or to form a "convex enveloppe" of the data to ensure good reconstruction [63, 65]. On the contrary, the templates learned through our approach seem to be located on a plane in the PCA space while the samples evolve orthogonally to this plane (when the FGW distance increases). In a classification context, this means that the learned templates will not actually represent realistic graphs from the data but might encode "exaggerated" or "extreme" features in order to maximize the margin between classes in the embedding. To reinforce this intuition, we added plots of the estimated templates in the supplementary. Finally, in the figure, the samples are coloured *w.r.t.* their class and the templates are coloured by their predicted class. Interestingly, the classes are already well separated with $4$ templates but the separation is clearly non-linear whereas using GNN pre-processing and a larger number of templates leads to a linear separation of the two classes.

**Sensitivity to the number of templates and GNN layers.** To illustrate the sensitivity of our TFGW layer to the number of templates K and the number of GNN layers L in $\phi_{\mathbf{u}}$, we learned our models on the PTC dataset with $L$ varying in $\{0, 1, 2, 3, 4\}$. We follow the same procedure than in the benchmark of Section 3.2 regarding the validation of $K$ and the learning process, while fixing the number of hidden units in the GNN layers to $16$. The test accuracy distributions for all settings are reported in Figure 5. Two phases are

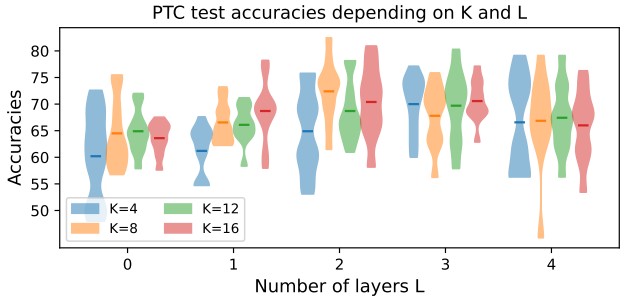

Figure 5: Test accuracy distributions by number of templates and number of GNN layers.

clearly distinguishable. The first one for $L \leq 2$, where for each $L$ we see that the performance across the number of templates steadily increases, and the second for $L > 2$ where this performance progressively decreases as a function of $L$. Moreover, in the first phase performances are considerably dependent on $K$ to compensate for a simple node representation, while this dependency is mitigated in the second which exhibits a slight overfitting. Note that these deeper models still lead to competitive results in comparison with benchmarked approaches in Table 2, with best averaged accuracies of $70.6$ ($L = 3$) and $67.4$ ($L = 4$). On one hand, these observations led us to set our number of layers to $L = 2$ for all benchmarked datasets which lead to strong generalization power. On the other hand, deeper models might be a way to benefit from our FGW embeddings with very few templates which can be interesting from a computational perspective on larger graph datasets.

## 4 Conclusion

We have introduced a new GNN layer whose goal is to represent a graph by its distances to template graphs, according to the optimal transport metric FGW. The proposed layer can be used directly on raw graph data as the first layer of a GNN or can also benefit from more involved node embedding using classical GNN layers. In a graph classification context, we combined this TFGW layer with a

simple MLP model. We demonstrated on several benchmark datasets that this approach compared favorably with state-of-the-art GNN and kernel based classifiers. A sensitivity analysis and an ablation study were presented to justify the choice of several parameters explaining the good generalization performances.

We believe that the new way to represent complex structured data provided by TFGW will open the door to novel and hopefully more interpretable GNN architectures. From a practical perspective, future works will be dedicated to combine TFGW with fast GPU solvers for network flow [51]. This would greatly accelerate our approach and more generally OT based deep learning methods. We also believe that the FGW distance and its existing extensions can be used with other learning strategies including semi-relaxed FGW [62] for sub-graph detection.

## 5 Acknowledgements

This work is partially funded through the projects OTTOPIA ANR20-CHIA-0030 and 3IA Côte d'Azur Investments ANR-19-P3IA-0002 of the French National Research Agency (ANR). This research was supported by 3rd Programme d'Investissements d'Avenir ANR-18-EUR-0006-02. This action benefited from the support of the Chair "Challenging Technology for Responsible Energy" led by l'X – Ecole polytechnique and the Fondation de l'Ecole polytechnique. This work is supported by the ACADEMICS grant of the IDEXLYON, project of the Université de Lyon, PIA operated by ANR-16-IDEX-0005. This project was supported in part by the AllegroAssai ANR project ANR-19-CHIA-0009. The authors are grateful to the OPAL infrastructure from Université Côte d'Azur for providing resources and support.

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
