# OpenReview forum: "Template based Graph Neural Network with Optimal Transport Distances"
_NeurIPS.cc/2022/Conference — NeurIPS 2022 Accept_

### Official Review · Reviewer_Md5Y · 2022-06-22

**Rating:** 6
**Confidence:** 4
**Soundness:** 3 good
**Presentation:** 3 good
**Contribution:** 2 fair

**Summary:**

The authors study the problem of graph classification. They propose to use the Fused Gromov-Wasserstein (FGW) distance as features for the MLP classifier for graph classification. In the proposed framework, instead of computing pair-wise FGW distances between each pair of graphs, they propose to create some learnable 'templates', which are expected to carry discriminating information either in the features or in the structure of the graphs. Calculating the FGW distance to the templates forms the TFGW layer. The authors also added a GNN layer at the beginning for graph feature transformations. The authors performed various experiments, ablation studies and visualizations to show the superior performance of the proposed framework.

**Questions:**

1. I don't quite understand the function of h here. Particularly, it seems the h is used to decide the coupling function. How does the initialization of h change the results? Can you do an ablation study where node weights are not in the framework to see what role it plays?

2. The use of GNN at the beginning: the author chosed GIN as the first layer. GIN itself is very discriminative (expressive) of the graph structure. If I understand correctly, the authors didn't use the pooling layer to get an embedding of the whole graph from GIN, but just updated the features of each node based on message propagation? The authors can clarify better on the process.

**Limitations:**

Would like to see a comparison of the computation time for the proposed method and other baselines too.

**Strengths And Weaknesses:**

The authors leveraged Fused Gromov Wasserstein distance for graph classification. The idea is intuitive, and the authors demonstrated the good performance of the model. However, some of the ideas presented can be better explained (see questions below)

---

> ### Author Response · Authors · 2022-08-02
> **Response to Reviewer Md5Y (1)**
>
> We sincerely appreciate your comments and suggestions. We made every effort to address all the concerns. In the following, we quote your comments and then give our detailed response point-by-point.
>
> > **Question 1**:  I don't quite understand the function of h here. Particularly, it seems the h is used to decide the coupling function. How does the initialization of h change the results? Can you do an ablation study where node weights are not in the framework to see what role it plays?
>
> We are not sure about which $\mathbf{h}$ you are referring to so we will discuss both $\mathbf{h}_i$ that weights the nodes of the input graphs and $\bar{\mathbf{h}}_k$ that weights the template nodes. Note that FGW always requires a weighting of the graph nodes to be introduced (however it can be set as uniform by default).
>
> In the experiments we have used the uniform weights $\mathbf{h}_i$ for all input graphs, which already led to state of the art performances across all datasets. We agree that other kind of node weights (e.g. normalized degree, or parameterized ones [65, 61]) could be employed. However, a full ablation study, in order to select the most effective weighting,  would require more parameters to be introduced and optimized. Also, notice that using uniform weights over the graph nodes is the standard procedure for OT based graph divergences [10,56,60,62,64].
>
> On the contrary, the weights $\bar{\mathbf{h}}_k$ of the templates are directly learned from the data. We actually did an ablation study with these weights (learnt or set to uniform in Table 4: $\bar{\mathbf{h}}_k$) and there is a consistent gain of $1\%-2\%$ on all datasets. This justifies the fact that we learn them. Note that learning the weights for templates/dictionaries have already been shown to lead to a more powerful modeling of the data in [62], in a much simpler linear model for graphs.
>
>
> > **Question 2**:  The use of GNN at the beginning: the author chosed GIN as the first layer. GIN itself is very discriminative (expressive) of the graph structure. If I understand correctly, the authors didn't use the pooling layer to get an embedding of the whole graph from GIN, but just updated the features of each node based on message propagation? The authors can clarify better on the process.
>
> Yes indeed we used only the GIN convolutional layers for $\phi_u$ and replaced the sum pooling in the traditional GIN architecture by the TFGW layer. This allows to have a fair comparison between the models by keeping the same GNN architecture on most methods.

---

> > ### Author Response · Authors · 2022-08-02
> > **Response to Reviewer Md5Y (2)**
> >
> > > **Limitations**:  Would like to see a comparison of the computation time for the proposed method and other baselines too.
> >
> > As illustrated in Table 3 TFGW has approximately the same or lower averaged prediction time per graph on CPU, as recent DropGIN and PPGN ${}^{(*)}$ architectures. We provide below (and will add to the paper) these practical runtimes on CPU and GPU for two datasets (PROTEIN has larger graphs in average than PTC)
> >
> > ${}^{(*)}$ : We harmonized our benchmark reported in Table 2 by evaluating PPGN on all datasets using a 10-fold CV with a holdout as for other methods. The main paper will be modified accordingly.
> >
> > - **On PTC dataset:**
> >
> > |      | CPU runtimes (ms) | GPU runtimes (ms) | Accuracy drop w.r.t TFGW(%)|
> > | :----: |:-----------------:|:-----------------:| :------------------------: |
> > | TFGW | 12.1 | 20.7 |  - |
> > | OT-GNN |7.6 | 8.8| 4.4|
> > |GIN|0.19|0.06|9.4|
> > |DropGIN|14.3|2.1|10.1|
> > |PPGN|32.1|26.1|6.8
> >
> > - **On PROTEIN dataset:**
> >
> > |      | CPU runtimes (ms) | GPU runtimes (ms) | Accuracy drop w.r.t TFGW(%)|
> > | :----: |:-----------------:|:-----------------:| :------------------------: |
> > | TFGW | 45.9 | 21.8 |  - |
> > | OT-GNN | 27.1 | 11.1| 4.9 |
> > |GIN|2.5|0.09|6.7|
> > |DropGIN|79.8|4.7|6.0|
> > |PPGN|91.7|11.6|5.8|
> >
> > We can see that GIN, DropGIN and PPGN, get a 3-30x speedup on GPU when TFGW is slower on GPU for PTC and 2x faster for PROTEIN (acceleration of matrix product for large graphs but overhead time for transfering between CPU and GPU for small graphs). This shows that TFGW is not yet accelerated on GPU but remains reasonable in practice and can still benefit from it on the GNN and MLP models.
> >
> > Note that the main bottleneck of the method is the OT network flow CPU solver that is called in the Conditional Gradient solver of FGW in the current implementation. Recent works [A,B,C], notably from NVIDIA, focused on accelerating network flow algorithms on GPU which will probably be integrated in CUDA and will allow a similar speedup for TFGW and GIN in the future. We strongly believe that the fact that the method is not yet optimized on GPU is only a temporary problem, similarly to  major contributions which  were notoriously slow in a recent past, when proposed, e.g MLP, CNN, LSTM, Attention networks etc.
> >
> > [A] Wu, J., He, Z., & Hong, B. (2012). Efficient CUDA Algorithms for the Maximum Network Flow Problem. In GPU Computing Gems Jade Edition (pp. 55-66). Morgan Kaufmann.
> >
> > [B] https://on-demand.gputechconf.com/gtc/2017/presentation/S7370-hugo-braun-efficient-maximum-flow_algorithm.pdf
> >
> > [C] https://mate.unipv.it/gualandi/talks/Gualandi_Aussois2020.pdf

---

> > > ### Comment · Reviewer_Md5Y · 2022-08-07
> > > **Response**
> > >
> > > Thanks to the authors for addressing the questions. I have read through the reviews and author's responses, and I remain my original rating of the paper.

---

### Official Review · Reviewer_KbbR · 2022-07-10

**Rating:** 7
**Confidence:** 3
**Soundness:** 3 good
**Presentation:** 4 excellent
**Contribution:** 3 good

**Summary:**

The authors introduce a novel GNN construction by adding a new GNN layer that is called TFGW for Template-based FGW. FGW is Fused Gromov-Wasserstein distance and it shows the similarity and isomorphism between two graphs. The novel GNN consists of normal GNN layers, a TFGW layer, and an MLP layer, and its structure is shown in Figure 1 of the paper. Furthermore, the properties and the invariances are discussed in the paper. It is illustrated that the TFGW embeddings are invariant to strong isomorphism. The optimization strategy and graph templates relevant for the downstream task are also introduced. The trade-off parameter is learned in the model in a supervised way which appears first. In section 3, the experiment settings and results are shown. The performance of the TFGW is tested in synthetic datasets, several well-known graph classification datasets, and an ablation study.  The results of the sensitivity analysis and an ablation study could show the generalization of the parameters in the model.

**Questions:**

It is reported in the paper that for L=0, the TFGW with SP matrices outperforms TFGW with ADJ matrices while for L no less than 1, the TFGW with ADJ matrices shows better performance. Could a more specific explanation be given? Is this phenomenon related to the aggregation function of the GNN itself?

From section 2 in the paper, we could see that the trade-off parameter and the templates are learned from the data. Therefore, different templates could appear with different parameters. A direct problem might be how to deal with this phenomenon?


**Limitations:**

From lemma 1 in this paper, we could know that for strong isomorphic graphs, the embeddings are invariant but there is not any description of the performance of the TFGW on weak isomorphic graphs.

In this paper, the expressiveness of the TFGW is not derived. There should be a WL-test to show the expressiveness of this novel GNN structure.

**Strengths And Weaknesses:**

The language is very fluent and the expression is most appropriate in this paper. The proof and the illustration of the experiment details are very clear. The structure of the Template-based FGW layer is very novel and creative. The FGW distance could express the similarity and isomorphism between two graphs and it appears to be appropriate to construct a layer deeply related to it. The results of the ablation study could also show the outstanding performance of this novel layer. This paper is the first to learn the trade-off parameter in a supervised way. It is not required to adjust the parameter to obtain the best performance for other users and could guarantee the generalization of the proposed Template-based FGW model. In the numerical study in section 3, the authors compare the proposed model with several widely used GNN models. From the results in table 2 and table 4, we could find that the performance of the proposed model is distinguished and could do well in the downstream task.

---

> ### Author Response · Authors · 2022-08-02
> **Response to Reviewer KbbR**
>
> We sincerely appreciate your comments and suggestions. We made every effort to address all the concerns. In the following, we quote your comments and then give our detailed response point-by-point.
>
> > **Question 1**:  It is reported in the paper that for L=0, the TFGW with SP matrices outperforms TFGW with ADJ matrices while for L no less than 1, the TFGW with ADJ matrices shows better performance. Could a more specific explanation be given? Is this phenomenon related to the aggregation function of the GNN itself?
>
> This is a very interesting question that is related to the expressiveness of SP and ADJ in our model. We believe that with few (L<=1) GNN Layers, SP, that contains global graph information, is more informative than ADJ that contains only local adjacency (binary) information, hence leads to better performances. When more layers are used, the GNN is able to capture this global information from the graph through the convolutions ($L$-hop neighborhood information aggregation for $L$ layers) and thus local structure (i.e. ADJ) becomes sufficient. This could also come from the fact that the ADJ matrices are normalized whereas the SP matrices are not (and their entries can increase linearly with the number of nodes!) which could be detrimental when enough global information is available in the features.
>
> In more general terms, the question of the input representations best suiting a given task is an opened question, actively studied in the context of GW/FGW (see [14,15,61,62] of the main paper). Our paper provides the very first insights on this matter in an end-to-end framework, considering the 2 first input graph representations suggested in the GW literature, ADJ and SP. As noted by the reviewer, these results motivate further research out of the scope of our paper and regarding several aspects, such as theoretical properties and/or relations with spectral GNN and spectral GW (see [15]).
>
>
>
>
>
> > **Question 2**:  From section 2 in the paper, we could see that the trade-off parameter and the templates are learned from the data. Therefore, different templates could appear with different parameters. A direct problem might be how to deal with this phenomenon?
>
> As noted by the reviewer, the $\alpha$ parameter is not a hyperparameter of the model but a parameter learned from the data. Therefore the templates and the $\alpha$ will eventually change w.r.t. different hyperparameters or initialization values, but as with CNN filters for instance, we do not consider it as being a problem, or do not see the reason why it should be one.
> One can observe that a scaling in the template features can be compensated by another value of $\alpha$, hence lead to identical FGW distances from inputs to templates. Is that the "problem" the reviewer is refering to? In practice, we observed that learning $\alpha$ as a parameter jointly with the templates leads to better performance than keeping it fixed and validating it as an hyperparameter.
>
> > **Limitation 1**:  From lemma 1 in this paper, we could know that for strong isomorphic graphs, the embeddings are invariant but there is not any description of the performance of the TFGW on weak isomorphic graphs.
>
> This is an interesting question but we think it is a bit out of the scope of the paper, as the concept of weak isomorphism between graphs intervene rarely in real-world graph classification datasets. Indeed, such invariance can occur when graphs have nodes which are exact duplicates from each other. For instance, it may occur with adjacency representation, when two nodes have selp-loops and are connected to all other nodes of the graph. These considerations can be found in the proofs of [14]. Moreover they were emphasized in [Theorem 1, Definition 1, A] which introduced a notion of “canonical representations” which amounts to merging the nodes which are exact duplicates and aggregate their respective masses.
>
> [A] Kerdoncuff, T., Emonet, R., Perrot, M., & Sebban, M. (2022, June). Optimal Tensor Transport. In Proceedings of the AAAI Conference on Artificial Intelligence (Vol. 36, No. 7, pp. 7124-7132).
>
>
> > **Limitation 2**:  In this paper, the expressiveness of the TFGW is not derived. There should be a WL-test to show the expressiveness of this novel GNN structure.
>
> We actually did something similar to a WL-test in Section 3.1 where we have illustrated in the simulated data that graphs that are indistinguishable by a WL-test can be discriminated perfectly with our method. This suggests that TFGW may be strictly more powerful, and further works will be dedicated to this analysis.

---

### Official Review · Reviewer_c2Hn · 2022-07-11

**Rating:** 8
**Confidence:** 3
**Soundness:** 3 good
**Presentation:** 3 good
**Contribution:** 3 good

**Summary:**

This work introduces TFGW, a novel graph embedding layer that describes a graph with its Fused-Gromov-Wasserstein distances from a set of learnt templates. TFGW is shown to exhibit strong discriminative power on a wide set of graph classification datasets.

**Questions:**

1. If my understanding is correct, the main novelty of TFGW in the context of OT-GNN is its use of the FGW distance instead of OT. In Lines 310-325, it is confirmed experimentally that the trade-off parameter $\alpha$ is not 0 or 1, so that the use of FGW is justified as it is not reduced back to OT. However, I'm wondering if an ablation study has been done to manually set $\alpha=0$ in TFGW, which reduces FGW back to OT, and compare its performance with that of learned $\alpha$? This should further cement the advantage of using FGW in place of OT, which I believe is an important point to make since FGW is more expensive to compute than OT.

2. How does the practical runtime for TFGW compare with other GNNs? I understand that the OT solver used is only implemented on CPU, and TFGW has a competitive speed when benchmarked on CPU. I am wondering if the current TFGW implementation is reasonably efficient on the experimental datasets.

3. The TFGW layer can serve as a drop-in for the global pooling step in any existing GNN model. However, it seems that TFGW is always used with GIN in this work. Have the authors tried TFGW with other GNN node embedding layers? Does TFGW interact better with some compared to others?

Misc.
1. Figure 1 can be seen from the web preview but cannot be displayed after the PDF file is downloaded and opened with Adobe. I am not sure if this is an issue on my side.

**Limitations:**

The authors have addressed the limitations of TFGW adequately.

**Strengths And Weaknesses:**

Overall, I found the paper to be clearly-written, well-motivated, and easy to follow. The experiments show meaningful improvements over existent baselines, and the sensitive analysis further substantiates the model's robustness. On the downside, there have been previous prototype-based graph embedding models (e.g. OT-GNN) that also learns the template in an end-to-end fashion, which somewhat limits the novelty of this work. In addition, I have some additional concerns that I hope the authors can clarify.

---

> ### Author Response · Authors · 2022-08-02
> **Response to Reviewer c2Hn (1)**
>
> We sincerely appreciate your comments and suggestions. We made every effort to address all the concerns. In the following, we quote your comments and then give our detailed response point-by-point.
>
> > **Question 1 :**  If my understanding is correct, the main novelty of TFGW in the context of OT-GNN is its use of the FGW distance instead of OT. In Lines 310-325, it is confirmed experimentally that the trade-off parameter $\alpha$  is not 0 or 1, so that the use of FGW is justified as it is not reduced back to OT. However, I'm wondering if an ablation study has been done to manually set $\alpha=0$ in TFGW, which reduces FGW back to OT, and compare its performance with that of learned ? This should further cement the advantage of using FGW in place of OT, which I believe is an important point to make since FGW is more expensive to compute than OT.
>
> **Answer - Question 1:**
>
> Yes an ablation has been performed for the case $\alpha=0$ and can be seen in the lines OT-GNN in Table 2 with $L=2$ and $L=4$ GIN layers. TFGW is clearly superior to OT-GNN on all datasets. As noted by the reveiwer the OT-GNN model is a special case or our TFGW model when fixing exactly $\alpha = 0$.
>
> We also want to emphasize that in addition to having far better performance than OT-GNN thanks to a better encoding (and learning) of the graph structure, TFGW has been evaluated on the task of graph classification whereas OT-GNN was only evaluated on few regression datasets.
>
> > **Question 2:** How does the practical runtime for TFGW compare with other GNNs? I understand that the OT solver used is only implemented on CPU, and TFGW has a competitive speed when benchmarked on CPU. I am wondering if the current TFGW implementation is reasonably efficient on the experimental datasets.
>
> **Answer - Question 2**:
>
> As illustrated in Table 3 TFGW has approximately the same or lower averaged prediction time per graph on CPU, as recent DropGIN and PPGN ${}^{(*)}$ architectures. We provide below (and will add to the paper) these practical runtimes on CPU and GPU for two datasets (PROTEIN has larger graphs in average than PTC).
>
>  ${}^{(*)}$: We harmonized our benchmark reported in Table 2 by evaluating PPGN on all datasets using a 10-fold CV with a holdout as for other methods. The main paper will be modified accordingly.
>
> - **On PTC dataset:**
>
> |      | CPU runtimes (ms) | GPU runtimes (ms) | Accuracy drop w.r.t TFGW(%)|
> | :----: |:-----------------:|:-----------------:| :------------------------: |
> | TFGW | 12.1 | 20.7 |  - |
> | OT-GNN |7.6 | 8.8| 4.4|
> |GIN|0.19|0.06|9.4|
> |DropGIN|14.3|2.1|10.1|
> |PPGN|32.1|26.1|6.8
>
> - **On PROTEIN dataset:**
>
> |      | CPU runtimes (ms) | GPU runtimes (ms) | Accuracy drop w.r.t TFGW(%)|
> | :----: |:-----------------:|:-----------------:| :------------------------: |
> | TFGW | 45.9 | 21.8 |  - |
> | OT-GNN | 27.1 | 11.1| 4.9 |
> |GIN|2.5|0.09|6.7|
> |DropGIN|79.8|4.7|6.0|
> |PPGN|91.7|11.6|5.8|
>
> We can see that GIN, DropGIN and PPGN, get a 3-30x speedup on GPU when TFGW is slower on GPU for PTC and 2x faster for PROTEIN (acceleration of matrix product for large graphs but overhead time for transfering between CPU and GPU for small graphs). This shows that TFGW is not yet accelerated on GPU but remains reasonable in practice and can still benefit from it on the GNN and MLP models.
>
> Note that the main bottleneck of the method is the OT network flow CPU solver that is called in the Conditional Gradient solver of FGW in the current implementation. Recent works [A,B,C], notably from NVIDIA, focused on accelerating network flow algorithms on GPU which will probably be integrated in CUDA and will allow a similar speedup for TFGW and GIN in the future. We strongly believe that the fact that the method is not yet optimized on GPU is only a temporary problem, similarly to  major contributions which  were notoriously slow in a recent past, when proposed, e.g MLP, CNN, LSTM, Attention networks etc.
>
> [A] Wu, J., He, Z., & Hong, B. (2012). Efficient CUDA Algorithms for the Maximum Network Flow Problem. In GPU Computing Gems Jade Edition (pp. 55-66). Morgan Kaufmann.
>
> [B] https://on-demand.gputechconf.com/gtc/2017/presentation/S7370-hugo-braun-efficient-maximum-flow_algorithm.pdf
>
> [C] https://mate.unipv.it/gualandi/talks/Gualandi_Aussois2020.pdf

---

> > ### Author Response · Authors · 2022-08-02
> > **Response to Reviewer c2Hn (2)**
> >
> > > **Question 3**: The TFGW layer can serve as a drop-in for the global pooling step in any existing GNN model. However, it seems that TFGW is always used with GIN in this work. Have the authors tried TFGW with other GNN node embedding layers? Does TFGW interact better with some compared to others?
> >
> > Indeed, we focused on this paper on GIN and performed a validation of its depth in an ablation study which led to state of the art performances on a number of well-known real-world datasets. GIN is one of the most popular and simple framework among most recent GNN and  provides injective mappings on the nodes that can preserve the discrimination capability of the TFGW layer.
> >
> > Due to lack of space we did not investigate other GNN architectures but the reviewer is right that TFGW can replace standard pooling layers in any other architecture. We cannot provide an extensive study before the response deadline, but we will complete our experiments with an alternative architecture (GAT [D]) and our TFGW layer, in the openreview forum as soon as possible and in the supplementary material for the final version of the paper.
> >
> >
> > [D] Veličković, Petar, et al. "Graph Attention Networks." International Conference on Learning Representations. 2018.
> > (reference which will be adapted to the main paper)
> >
> >
> > > **Misc**: Figure 1 can be seen from the web preview but cannot be displayed after the PDF file is downloaded and opened with Adobe. I am not sure if this is an issue on my side.
> >
> > We are sorry about this inconveniency but we could not reproduce the described issue with several PDF readers including our Adobe versions. So it is most likely a configuration problem on your side.

---

> > > ### Author Response · Authors · 2022-08-08
> > > **Response to Reviewer c2Hn (3)**
> > >
> > > **Complementary results for question 3**:
> > >
> > > As promised, we conducted a study using Graph Attention Networks (GAT) [D] instead of GIN. The main conclusions are:
> > >
> > > - Our TFGW layer to obtain the graph representations consistently leads to better performances than simple sum pooling over graph nodes, regardless of the GNN architectures.
> > > - Performances of both approaches seem correlated.
> > >
> > > We now detail our study which will be added to the supplementary material.
> > >
> > > First, as the GAT architecture was investigated by its authors on node classifications and not graph classifications, we study on 3  datasets the behavior of GAT layers within the same framework than the one proposed by GIN (detailed in section 3 of the supplementary), where the number of GAT layers L is chosen in {1,2,3,4} . The hyperparameters are also validated in an analog way, while using a single attention head in GAT layers. The results in terms of accuracy are reported in the following table, where the best model (resp. second best) is highlighted in bold (rest. italic):
> > >
> > >
> > >
> > > |           | MUTAG         | PTC           |    PROTEIN    |
> > > | --------- | ------------- | ------------- |:-------------:|
> > > | GAT (L=1) | 89.4(1.0)     | 53.1(3.4)     | **77.8(1.7)** |
> > > | GAT (L=2) | *91.1(2.5)*     | 52.0(4.0)     |   76.3(3.1)   |
> > > | GAT (L=3) | 88.9(1.7)     | *53.4(3.8)* |   75.9(2.3)   |
> > > |   GAT (L=4) | **91.2(2.8)** | 50.9(5.8)     |   *77.6(2.7)*   |
> > > | GIN | 90.1(4.4) | **63.1(3.9)**     |   76.2(2.8)   |
> > >
> > > GAT provides competitive performances on MUTAG and PROTEIN compared to GIN, while GIN largely outperforms GAT on PTC. Also, it was harder to find a consensus across datasets on L for GAT-based architectures compared to GIN's ones. Finally, we observed that using multi-head attentions was prone to overfitting and led to lower performances on these graph classification tasks, so such suggestions from GAT's authors on node classification tasks seem to not hold for these graph-level tasks.
> > >
> > > Finally, we investigate the merge of our TFGW layer and GAT, following an analog validation than for TFGW coupled with GIN, setting L in {1, 2}. The results are reported in the following table, where the best model (resp. second best) is highlighted in bold (rest. italic):
> > >
> > >
> > >
> > > |                     | MUTAG | PTC | PROTEIN |
> > > | ------------------- | ----- | --- | ------- |
> > > | TFGW ADJ (GAT, L=2) |   95.4(3.5)|  68.7(5.8)  | **83.4(2.8)**|
> > > | TFGW SP (GAT, L=2)  |  *96.2(3.0)* |  67.9(5.8) |  82.6(2.9) |
> > > | TFGW ADJ (GAT, L=1) |  94.8(3.1) |  66.9(5.4)  |  82.1(3.3)    |
> > > | TFGW SP (GAT, L=1)  |   **96.4(3.3)**    |  68.3(6.0)   |  82.3(3.1)       |
> > > | TFGW ADJ (GIN, L=2) |      **96.4(3.3)**|**72.4(5.7)**|*82.9(2.7)*|
> > > | TFGW SP (GIN, L=2)  |     94.8(3.5)|70.8(6.3)|82.0(3.0)|
> > > | TFGW ADJ (GIN, L=1) |      94.8(3.1) |68.7(5.8)|81.5(2.8)|
> > > | TFGW SP (GIN, L=1)  |    95.4(3.5)   | *70.9(5.5)*  |  82.1(3.4)  |
> > >
> > > We can see that TFGW with GAT leads to competitive performances compared to TFGW with GIN, at least of MUTAG and PROTEIN. GAT clearly struggles on the PTC dataset, however with our TFGW layer instead of a sum pooling, we observe a boost of performances from 53.4% to 68.7%. Even if TFGW coupled with GIN on PTC is still considerably better than TFGW with GAT. Therefore, the choice of GNN architectures to produce node embeddings to feed to the TFGW layer matters, but the gain from using TFGW seems to be independent of the GNN architecture.

---

> > > > ### Comment · Reviewer_c2Hn · 2022-08-08
> > > > **Response to authors**
> > > >
> > > > Thank you for the detailed discussion. Overall, my concerns have been addressed adequately. The runtime increase of TFGW seems acceptable compared to its performance gain, and I agree that the framework can benefit greatly from GPU support in the future. The results on GAT also look encouraging, and I think the important point is that TFGW improves over traditional pooling methods. In light of the above, I have raised my score to 8.

---

### Meta-Review · Area_Chair_Cytd · 2022-08-31

**Recommendation:** Accept
**Confidence:** Certain

**Metareview:**

In this paper, the authors introduced a new GNN layer to represent a graph by the distances to template graphs. hey used the OT(FGW, used-Gromov-Wasserstein) distance as the metric. They showed good performance on several benchmark datasets.  Overall, the paper is very well written and motivated. The description of the methods and the presentation of the experiments are convincing. We also thank the careful and detailed rebuttal from the authors. This is a pleasant read paper.


**Award:**

No

---

### Decision · Program_Chairs · 2022-09-14

Accept